# Levosimendan as a “Bridge to Optimization” in Patients with Advanced Heart Failure with Reduced Ejection—A Single-Center Study

**DOI:** 10.3390/jcm11144227

**Published:** 2022-07-21

**Authors:** Daniele Masarone, Michelle M. Kittleson, Maria L. Martucci, Fabio Valente, Rita Gravino, Marina Verrengia, Ernesto Ammendola, Carla Contaldi, Vito Di Palma, Angelo Caiazzo, Andrea Petraio, Piero Pollesello, Giuseppe Pacileo

**Affiliations:** 1Heart Failure Unit, Department of Cardiology, AORN dei Colli-Monaldi Hospital Naples, 80131 Naples, Italy; marialuigia.martucci@ospedalideicolli.it (M.L.M.); fabio.valente@ospedalideicolli.it (F.V.); rita.gravino@ospedalideicolli.it (R.G.); marina.verrengia@ospedalideicolli.it (M.V.); ernesto.ammendola@ospedalideicolli.it (E.A.); carla.contaldi@ospedalideicolli.it (C.C.); vito.dipalma@ospedalideicolli.it (V.D.P.); giuseppe.pacileo@ospedalideicolli.it (G.P.); 2Department of Cardiology, Smidt Heart Institute, Cedars-Sinai, Los Angeles, CA 90048, USA; michelle.kittleson@cshs.org; 3Heart Transplant Unit, Department of Cardiac Surgery and Transplant, AORN dei Colli-Monaldi Hospital, 80131 Naples, Italy; angelo.caiazzo@ospedalideicolli.it (A.C.); andrea.petraio@ospedalideicolli.it (A.P.); 4Critical Care, Orion Pharma, 02101 Espoo, Finland; piero.pollesello@orionpharma.it

**Keywords:** levosimendan, disease modifier drugs, advanced heart failure, heart failure reduced ejection fraction

## Abstract

*Background*: Patients with advanced heart failure with reduced ejection fraction often cannot tolerate target doses of guideline-directed medical therapy due to symptomatic hypotension, renal dysfunction, and associated electrolyte abnormalities. While levosimendan can facilitate the titration of β-blockers in patients with advanced HFrEF, it is unclear whether ambulatory levosimendan infusions would offer the same benefit. In this prospective study, we investigate the effects of intermittent ambulatory levosimendan infusions on the uptitration of disease-modifying drugs. *Methods*: We enrolled 37 patients with advanced HFrEF who received repeated ambulatory infusions of levosimendan between January 2018 and January 2021. The demographic, clinical, and laboratory data were acquired 24 h before the first and the last ambulatory levosimendan infusion. *Results*: At the 1 year follow-up, the enrolled patients were on significantly higher doses of guideline-directed medical therapy, including bisoprolol (3.2 ± 2.8 mg vs. 5.9 ± 4.1 mg; *p* = 0.02), sacubitril/valsartan (41.67 ± 32.48 mg vs. 68.5 ± 35.72 mg; *p* = 0.01), and eplerenone (12.7 ± 8.5 mg vs. 22.8 ± 13.6 mg; *p* = 0.03). Furthermore, a substantial decrease in the furosemide dose was observed (123.2 ± 32.48 mg vs. 81.6 ± 19.47 mg; *p* < 0.0001). *Conclusions*: Levosimendan facilitates the optimization of disease-modifying heart failure medications in previously intolerant advanced HFrEF patients.

## 1. Introduction

Despite improvements in pharmacological and nonpharmacological treatments for patients with heart failure (HF) with reduced ejection fraction (HFrEF), approximately 10% of patients have a progressively worsening functional status culminating in advanced HF [1]. Furthermore, patients with advanced HFrEF develop distinct haemodynamic features that affect their natural history and disease-modifying drugs tolerance [2]. Symptomatic hypotension, renal dysfunction, and hyperkalaemia render the uptitration of β-blockers, angiotensin receptor-neprilysin inhibitors (ARNIs), and mineral receptor antagonists (MRAs) challenging [3]. Levosimendan is a calcium-sensitising medication [4] with two mechanisms of action, increased inotropy and vasodilation, and positive haemodynamic effects in acute HF [5]. Several studies of levosimendan in advanced HFrEF have been performed; however, they all included a bolus dose mimicking acute treatment [6,7]. More recently, the LIONHEART study showed that ambulatory intermittent levosimendan infusions reduced NT-proBNP plasma levels and hospitalisations [8]. Following this pivotal trial, subsequent studies demonstrated that intermittent ambulatory infusions of levosimendan improved haemodynamic parameters [9] and functional capacity [10], while reducing hospitalisation [11,12] in patients with advanced HFrEF. In addition, a 24-h infusion of levosimendan could facilitate the titration of β-blockers in previously intolerant advanced HFrEF patients [13]. However, the role of levosimendan ambulatory infusions in the optimization of guideline-directed medical therapy for HFrEF remains unknown. Therefore, the purpose of this prospective study was to investigate whether intermittent infusions of levosimendan could facilitate the titration of β-blockers, ARNIs, and MRAs in advanced patients with HFrEF and a documented intolerance to disease-modifying drugs uptitration.

## 2. Materials and Methods

### 2.1. Study Population

We enrolled the study population at the Heart Failure Unit of Monaldi Hospital between January 2018 and January 2021 (Figure 1).

The following inclusion criteria were used:(1)HFrEF with a left ventricular ejection fraction <35%,(2)NYHA class III-IV,(3)NT-proBNP >2500 pg/mL,(4)walking distance at 6-min walking test <300 m,(5)indication for intermittent ambulatory levosimendan infusion due to episodes of pulmonary or systemic congestion requiring a high dose i.v. diuretics or episodes of low output requiring inotropes or causing >2 unplanned visits or hospitalisations in the last 12 months, and(6)guideline-directed medical therapy for HFrEF not at target dose [14,15,16], with documented intolerance to their uptitration in the six months prior to levosimendan infusion.

The following exclusion criteria were used:(1)End-stage renal disease (i.e., estimated glomerular filtration rate <15 mL/kg/min according to the CKD-EPI equation),(2)severe liver impairment (i.e., Child–Pugh score >10).

Signed informed consent was obtained, the Declaration of Helsinki was followed, and the institutional review board of AORN dei Colli–Ospedale Monaldi granted approval (deliberation n° 345 of November 2017). Demographic, clinical, and laboratory data were acquired from stable patients 24 h before the first and the last ambulatory levosimendan administration. The patients were followed up for 1 year during ambulatory infusions of levosimendan, and the follow-up was started at the first infusion of levosimendan.

### 2.2. Levosimendan Infusion

In all patients, levosimendan (Simdax^®^) was intravenously administered at 0.2 µg/kg/min for a total dosage of 6.25 mg every two weeks in an ambulatory setting. Levosimendan was administered in all patients for at least 1 year. No change in the dose of levosimendan occurred during the follow-up.

### 2.3. Evaluation of Disease Modifiers Drug Dose

During follow-up, the doses of disease-modifying drugs were uptitrated according to clinical judgment by two physicians with experience treating patients with advanced HFrEF (D.M., F.V.). The uptitration of the drugs was performed in an ambulatory setting on the same day as the levosimendan infusion. The doses of guideline-directed medical therapy were recorded 24 h before the first and the last ambulatory infusion of levosimendan; the latter doses were considered the maximum doses for each patient.

### 2.4. Statistical Analysis

All statistical analyses were performed using Prism 9 (GraphPad Software, San Diego, CA, USA). All demographic and clinical variables are expressed as the mean ± standard deviation. Categorical variables are expressed as numbers and percentages. Differences between the baseline and treatment values were compared using a Wilcoxon rank test for non-normal distribution and using a *t*-test for normal distribution. All *p*-values were two-sided; *p* < 0.05 indicated statistical significance.

## 3. Results

A total of 71 patients meeting the diagnostic criteria for advanced HFrEF with an indication for intermittent infusion of levosimendan were screened in our unit during the study period. Of these patients, fifteen (21%) did not receive an ambulatory infusion of levosimendan for end-stage renal disease, and five patients (7%) did not for severe liver failure. In addition, six patients (8%) had HF-related hospitalisations in the month before levosimendan administration, and eight patients (11%) had already achieved the target dose of disease-modifying drugs, so they were excluded from the study. The final population comprised 37 patients (mean age 55.8 ± 13.2 years, 84% male, mean ejection fraction 26.8 ± 9.4%). The demographic, clinical, and echocardiographic characteristics of the study population are presented in Table 1.

At the one-year follow-up, the ambulatory infusion of levosimendan had allowed a significant increase in the mean dose of sacubitril/valsartan compared with the dose before levosimendan treatment (41.67 ± 32.48 mg vs. 68.5 ± 35.72 mg; *p* = 0.01; Figure 2A).

Likewise, we observed a significant increase in the mean dose of bisoprolol compared with the dose before levosimendan administration (3.2 ± 2.8 mg vs. 5.9 ± 4.1 mg; *p* = 0.02; Figure 2B), and the same change was seen with eplerenone (12.7 ± 8.5 mg vs. 22.8 ± 13.6 mg, *p* = 0.03; Figure 2C). Simultaneously with the increase in the dose of disease-modifying drugs, a substantial decrease in the dose of furosemide was observed compared with the dose before levosimendan treatment (123.2 ± 32.48 mg vs. 81.6 ± 19.47 mg; *p* < 0.0001; Figure 2D).

## 4. Discussion

One of the most complex clinical challenges in patients with advanced HFrEF is their intolerance to guideline-directed medical therapy or, if administered, inability to titrate to recommended doses due to hypotension, renal failure, and hyperkalaemia [17,18]. The poor tolerance of neurohormonal modulatory drugs in patients with advanced HFrEF could be related to the progression of the disease itself, leading to a critical reduction in the stroke volume resulting in hypotension and renal dysfunction. Alternatively, it could be associated with the direct effect of neurohormonal modulators or a combination of both [19]. Regardless of the cause, suboptimal doses of guideline-directed medical therapy in patients with advanced HFrEF are associated with poor prognoses. In this clinical scenario, levosimendan can assist in optimising therapy with β-blockers and drugs interfering with the renin-angiotensin-aldosterone system. Berger and colleagues demonstrated that levosimendan allowed the uptitration of β-blockers in previously intolerant HF patients. Levosimendan was periodically infused every four weeks, with a loading dose of 12 μg/kg for 10 min and an infusion rate of 0.1 μg/kg/min for 24 h. This protocol allowed for an increased dose of bisoprolol in patients in whom this had not been previously possible [13]. In our study, the use of levosimendan allowed for an increase in the dose of bisoprolol; this may have been due to the increase in cardiac output and consequent increase in blood pressure. The ability of levosimendan to increase cardiac output and cardiac performance, in addition to its positive effect on renal haemodynamics [20,21], may have allowed the uptitration of sacubitril/valsartan. Additionally, the positive impact on the renal performance, the reduction in the diuretic dose, and the reduction in potassium levels associated with levosimendan may have allowed the increase in the dose of MRAs [22]. Finally, in our study as well as in clinical trials [23] and in previous real-world experiences [24], the increasing dose of sacubitril/valsartan reduced the relative need for diuretics in patients with advanced HFrEF. This is potentially related to the natriuretic effects of sacubitril [25] or the presumed improvement in renal haemodynamics that may occur with sacubitril/valsartan [26].

## 5. Study Limitations

We recognise that the relatively small sample size, single-centre study design, the study’s observational nature, and the absence of a control arm could have affected our results. However, the data from our observational study should be taken into consideration when planning properly powered randomised clinical trials in this therapeutic setting.

## 6. Conclusions

Levosimendan facilitates the optimisation of guideline-directed medical therapy in patients with advanced HFrEF who were previously unable to achieve target doses. This therapeutic strategy may be used as a ‘bridge to optimisation’ and may justify, at least in part, the improvement in clinical outcomes that the intermittent infusion of levosimendan produces in patients with advanced HFrEF.

## Figures and Tables

**Figure 1 jcm-11-04227-f001:**
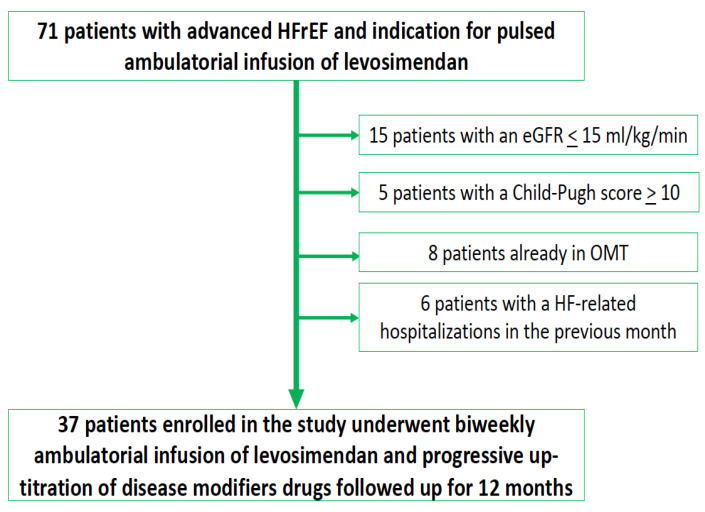
Study protocol. eGFR: estimated glomerular filtration rate. OMT: optimal medical therapy. HF: Heart failure. HFrEF: heart failure with reduced ejection fraction.

**Figure 2 jcm-11-04227-f002:**
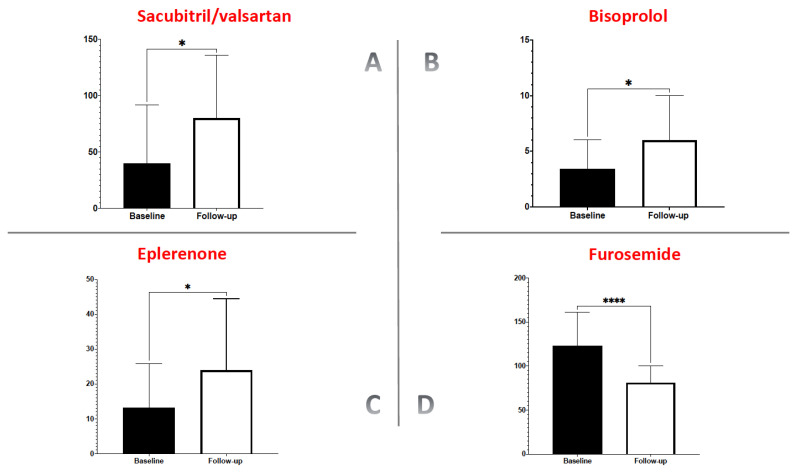
Change in the dose of disease modifier drugs (panel **A**–**C**) and diuretics at follow-up (panel **D**). *: *p* < 0.05; ****: *p* < 0.0001.

**Table 1 jcm-11-04227-t001:** Baseline clinical and echocardiographic characteristics of the study population.

Variable	Total Population (*n* = 37)
Age (mean ± SD)	55.8 ± 13.2 years
Female sex (*n*, %)	6 (16%)
Ischaemic (*n*, %)	20 (54%)
Hypertension (*n*, %)	18 (48%)
Diabetes (*n*, %)	17 (45%)
COPD (*n*, %)	12 (32%)
NYHA class III (*n*, %)	25 (67%)
NYHA class IV (*n*, %)	12 (33%)
SBP (mean ± SD)	97 ± 10 mmHg
DBP (mean ± SD)	62 ± 8 mmHg
NT-pro BNP (mean ± SD)	3448 ± 1187 pg/mL
Atrial fibrillation	15 (40%)
Hb (mean ± SD)	11.7 ± 1.8 g/dL
Creatinine (mean ± SD)	1.4 ± 1.3 mg/dL
eGFR (mean ± SD)	36.7 ± 18.1 mL/min/1.73 m^2^
LVEDV (mean ± SD)	2321.2 ± 85.9 mL
LVESV (mean ± SD)	192.7 ± 80.2 mL
LVEF (mean ± SD)	26.8 ± 9.4%
E wave (mean ± SD)	128.1 ± 39.5 cm/s
e’ average (mean ± SD)	6.9 ± 3.5 cm/s
E/e’ average (mean ± SD)	21.2 ± 6.3
DecT (mean ± SD)	165.2 ± 28.3 m/s
LAVi (mean ± SD)	52.5 ± 13.5 mL/m^2^
PASP (mean ± SD)	40.8 ± 12.6 mmHg
TAPSE (mean ± SD)	14.1 ± 5.4 mm
Peak systolic s wave (mean ± SD)	8.7 ± 3.2 cm/s
Loop diuretic (*n*, %)	37 (100%)
Furosemide dose (mean ± SD)	123.2 ± 32.48 mg
β-blocker (*n*, %)	37 (100 %)
Bisoprolol dose (mean ± SD)	3.2 ± 2.8 mg
ARNI (*n*, %)	37 (100%)
ARNI dose (mean ± SD)	41.67 ± 32.48 mg
MRA (*n*, %)	37 (100%)
Eplerenone dose	9.7 ± 8.8 mg

COPD: chronic obstructive pulmonary disease; NYHA: New York Health Association; SBP: systolic blood pressure; DBP: diastolic blood pressure; NT-pro BNP: N terminal-pro brain natriuretic peptide; Hb: haemoglobin; eGFR: estimated glomerular filtration rate; LVEDV: left ventricular end-diastolic volume; LVESV: left ventricular end-systolic volume; LVEF: left ventricular ejection fraction; E wave: peak early mitral inflow velocity; e′ average: average of septal and lateral peak early diastolic mitral annular velocity; DecT: deceleration time; LAVi, left atrium volume index; PASP: pulmonary artery systolic pressure; TAPSE: tricuspid annular plane systolic excursion; ARNI: angiotensin receptor-neprilysin inhibitor; MRA: mineral receptor antagonist.

## Data Availability

The data presented in this study are available on request from the corresponding author.

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
