# Peer review of "Levosimendan as a “Bridge to Optimization” in Patients with Advanced Heart Failure with Reduced Ejection—A Single-Center Study"

_jcm, 2022, doi:10.3390/jcm11144227_

Round 1

Reviewer 1 Report

Masarone et al. investigate in in their article – Levosimendan as a “bridge to optimization” in patients with advanced heart failure with reduced ejection – A single center study – the effect of ambulantory pulsed infusions of Levosimendan in 36 patients on the up-titration of heart failure medications Bisoprolol, Epleneron as wells as Sacubitril/Valsartan. They were able to demonstrate a significant increase in the dosage of these drugs.

In general, the authors address an interesting topic, the ambulatory optimization HErEF. They used a prospective, nonblinded approach with no control group. Patients were included over a 3-year period.

The study has the following weaknesses:

1.      It is unclear how many patients were screened for the study during the 3-year period, how many of the screened patients were not included, and why.

2.       The included patients then received 6.25 mg of Levosimendan every 2 weeks. It is not explained how long they received Levosimendan. It is also unclear what the target dose was for the patients and whether all included patients received this target dose and if not, why they did not receive it.

3.       The start of follow-up is undefined and unclear.

4.       The end of follow-up or the last follow-up is undefined and unclear.

5.       The authors did not address when the first and last time point of up-titration was in terms of Levosimendan dose and whether there were differences in Levosimendan dose when up-titration began.

6.       Time dependent up-titration of disease modifier drugs regarding Levosimendan dose is missing.

7.       Control group is missing

8.       Line 65 states, "During follow-up, the doses of disease modifier were up-titrated…". Unfortunately, Masarone et al. did not define the start of follow-up.

9.       Based on the text, the reader assumes that follow-up occurs after the pulsed Levosimendan infusions. However, from lines 67 and 68, it appears that the up-titration of the disease-modifying drugs occurred during the Levosimendan infusion.

10.   Lines 78/79 state that nine patients were excluded because of the maximum dose of drugs affecting the disease. It is unclear why these patients were included in the study and why they received at least one dose of Levosimendan. Given the brief study protocol, these patients should have been included in the first place.

11.   Line 80 states “The finale population comprised 36 patients…” but in table 1 the overall population as well as the percentage numbers are calculated for 37 patients.

12.   Figure 1-3 could be combined into one Figure.

13.   Figure legends differ from the text and are not consistent.

14.   For an original article, this manuscript contains little content.

In summary, Masarone and colleagues address an interesting topic, but the design and confusion of the study presented is unable to answer the question accurately.

Author Response

Masarone et al. investigate in in their article – Levosimendan as a “bridge to optimization” in patients with advanced heart failure with reduced ejection – A single center study – the effect of ambulantory pulsed infusions of Levosimendan in 36 patients on the up-titration of heart failure medications Bisoprolol, Epleneron as wells as Sacubitril/Valsartan. They were able to demonstrate a significant increase in the dosage of these drugs.

In general, the authors address an interesting topic, the ambulatory optimization HErEF. They used a prospective, nonblinded approach with no control group. Patients were included over a 3-year period.

Response: We thanks the reviewer for their comment on the topic of the paper

The study has the following weaknesses:

  1. It is unclear how many patients were screened for the study during the 3-year period, how many of the screened patients were not included, and why.

Response: We thank to reviewer for their comment. We have added in the section 2.1 how many patients were screened in the study period as well as the inclusion and exclusion criteria of the study. Also, in the same section we have add a Figure (Figure 1) with the study protocol.

  1. The included patients then received 6.25 mg of Levosimendan every 2 weeks. It is not explained how long they received Levosimendan. It is also unclear what the target dose was for the patients and whether all included patients received this target dose and if not, why they did not receive it.

Response:  We thank to reviewer for their comment. We have added in the section 2.2 that all patients received the dose of 625 mg at the target dose of 0.2 µg/kg/min biweekly

  1. The start of follow-up is undefined and unclear.

Response:  We thank to reviewer for their comment. We have added in the section 21 the start of follow-up.

  1. The end of follow-up or the last follow-up is undefined and unclear.

Response: We thank to reviewer for their comment. We have added in the section 2.1 that the patients were followed-up for 1 year.

  1. The authors did not address when the first and last time point of up-titration was in terms of Levosimendan dose and whether there were differences in Levosimendan dose when up-titration began.

Response: We thank to reviewer for their comment. We have clarified in the section 2.2 that that no changes in levosimendan doses occurred during follow-up

  1. Time dependent up-titration of disease modifier drugs regarding Levosimendan dose is missing

Response: We thank to reviewer for their comment. The dose of disease modifying drugs were increased during the study not at prespecified intervals but according to clinical judgment. So, unfortunately, we are not able to provide a time dependent uptitration of disease modifier drugs regarding Levosimendan.

  1. Control group is missing

Response: We agree with reviewer that the lack of control group is a limitation of the study, however due to ethical reasons a control group is not included in the study protocol.

  1. Line 65 states, "During follow-up, the doses of disease modifier were up-titrated…". Unfortunately, Masarone et al. did not define the start of follow-up.

Response: We thank to reviewer for their comment. In the section 2.1 we have clarify that the follow-up was started at the first infusion of levosimendan.

  1. Based on the text, the reader assumes that follow-up occurs after the pulsed Levosimendan infusions. However, from lines 67 and 68, it appears that the up-titration of the disease-modifying drugs occurred during the Levosimendan infusion.

Response: We thank to reviewer for their comment. In the section 2.1and 2.3 we have clarified the start of the follow-up and the time of disease modifying drugs uptitration.

  1. Lines 78/79 state that nine patients were excluded because of the maximum dose of drugs affecting the disease. It is unclear why these patients were included in the study and why they received at least one dose of Levosimendan. Given the brief study protocol, these patients should have been included in the first place.

Response: We thank to reviewer for their comment. In the section 2.1 we have clarified that one of the inclusion criteria was “disease modifiers drugs not at target dose, with documented intolerance to their up-titration in the six months prior to levosimendan infusion”

  1. Line 80 states “The finale population comprised 36 patients…” but in table 1 the overall population as well as the percentage numbers are calculated for 37 patients.

Response: We thank to reviewer for their comment that allow us the possibility to correct a clear mistake. in the new version of the manuscript, we have clarify that the final population comprised 37 patients.

  1. Figure 1-3 could be combined into one Figure

Response: We thank to reviewer for their comment. In the new version of the manuscript, we have combined Figure 1, Figure 2 and Figure 3 in a single Figure (Figure 2).

  1. Figure legends differ from the text and are not consistent.

Response: We thank to reviewer for their comment. We have carefully revised Figure legends

  1. For an original article, this manuscript contains little content

Response: We thank to reviewer for their comment.  We have improved the manuscript according to the suggestions of the reviewer

In summary, Masarone and colleagues address an interesting topic, but the design and confusion of the study presented is unable to answer the question accurately.

Response: Thank again to the reviewer for their positive and constructive comments on the manuscript. Based on this comment we have extensively revised the study protocol and the conduction of the study with the help of a new co-author with great expertise in the field of advanced heart failure and of medical research (Michelle M. Kittleson). We hope that the new version of the manuscript will be appreciated by the reviewer.

Reviewer 2 Report

The paper demonstrates that levosimendan improves heart failure drug tolerance. The authors presented increased doses of heart failure drugs at the beginning and end of follow up period.

I have several issues with the design of study.

Comment 1:  I believe that the study fails to demonstrate that levosimendan plays a role in increasing drug tolerability because there is no control group. In a way, patients without the use of levosimendan may also manage to reach maximal dose over the same period. In order to demonstrate the effect of levosimendan, the authors should consider presenting a control group without the use of levosimendan and compare two groups bypropensity score matching or multivariable adjustment.

Comment 2: I think there is insufficient information provided to the reader of the role of levosimendan in the introduction. They may improve the manuscript by briefly describing the mechanism of action of the drug and the previously demonstrated RCT results, as well as the limitation the drug has. 

Comment 3: The manuscript need English polishing, especially in the abstract.

Author Response

The paper demonstrates that levosimendan improves heart failure drug tolerance. The authors presented increased doses of heart failure drugs at the beginning and end of follow up period.

I have several issues with the design of study.

Comment 1: I believe that the study fails to demonstrate that levosimendan plays a role in increasing drug tolerability because there is no control group. In a way, patients without the use of levosimendan may also manage to reach maximal dose over the same period. In order to demonstrate the effect of levosimendan, the authors should consider presenting a control group without the use of levosimendan and compare two groups bypropensity score matching or multivariable adjustment.

Response: We agree with reviewer that the lack of control group is a limitation of the study, however due to ethical reasons a control group is not included in the study protocol. However, the patients enrolled in the study are stable patients with a previous documented intolerance to uptitration od disease modifying drugs in the previous six months, we think that every patient could a control of himself. 

Comment 2: I think there is insufficient information provided to the reader of the role of levosimendan in the introduction. They may improve the manuscript by briefly describing the mechanism of action of the drug and the previously demonstrated RCT results, as well as the limitation the drug has. 

Response: We thanks to the reviewer for their comment. In the new version of the manuscript, we have provided more details on the mechanism of action of levosimendan  and on their role in the management of patients with advanced HFrEF.

Comment 3: The manuscript need English polishing, especially in the abstract.

Response: We thanks to the reviewer for their comment.  We have extensively revised the manuscript with the help of  a native English-speaking co-author with great expertise in advanced heart failure and medical research (Michelle M. Kittleson).

Round 2

Reviewer 1 Report

Masarone et al. investigate in in their article – Levosimendan as a “bridge to optimization” in patients with advanced heart failure with reduced ejection – A single center study – the effect after a 1-year ambulantory pulsed infusions of Levosimendan in 37 patients on the up-titration of heart failure medications Bisoprolol, Epleneron as wells as Sacubitril/Valsartan and the reduction of Furosemide. They were able to demonstrate a significant increase and decrease in the dosage of these drugs, respectively.

In general, the authors address an interesting topic, the ambulatory optimization HErEF. They used a prospective, nonblinded approach with no control group. Patients were included over a 3-year period and the follow-up was collected after 1 year.

The authors improved significantly the structure and presentation of their article. They now present a clear screening, inclusion and exclusion procedure with a defined follow-up. In addition, the results, discussion, and conclusion are clearly presented.

As a commentary not relevant to the revision, it would have been interesting and would improve the strength of the study if the authors had also presented clinical correlates, such as improvement in EF in TTE or NYHA classification and laboratory parameters like proBNP.

Reviewer 2 Report

I have no further comments.